# Genome Editing in Zebrafish by ScCas9 Recognizing NNG PAM

**DOI:** 10.3390/cells10082099

**Published:** 2021-08-16

**Authors:** Yunxing Liu, Fang Liang, Zijiong Dong, Song Li, Jianmin Ye, Wei Qin

**Affiliations:** 1State Key Laboratory of Chemical Oncogenomics, Key Laboratory of Chemical Genomics, Peking University Shenzhen Graduate School, Shenzhen 518055, China; 1901111879@pku.edu.cn; 2Guangdong Provincial Key Laboratory for Healthy and Safe Aquaculture, Guangdong Provincial Engineering Technology Research Center for Environmentally-Friendly Aquaculture, Institute of Modern Aquaculture Science and Engineering, School of Life Sciences, South China Normal University, Guangzhou 510631, China; rachel8l@m.scnu.edu.cn (F.L.); 2019022509@m.scnu.edu.cn (Z.D.); 3Key Laboratory of Brain, Cognition and Education Sciences, Ministry of Education, Institute for Brain Research and Rehabilitation, South China Normal University, Guangzhou 510631, China

**Keywords:** CRISPR/Cas9, ScCas9, gene editing, zebrafish, NNG PAM

## Abstract

The CRISPR/Cas9 system has been widely used for gene editing in zebrafish. However, the required NGG protospacer adjacent motif (PAM) of *Streptococcus pyogenes* Cas9 (SpCas9) notably restricts the editable range of the zebrafish genome. Recently, Cas9 from *S. canis* (ScCas9), which has a more relaxed 5′-NNG-3′ PAM, was reported to have activities in human cells and plants. However, the editing ability of ScCas9 has not been tested in zebrafish. Here we characterized and optimized the activity of ScCas9 in zebrafish. Delivered as a ribonucleoprotein complex, ScCas9 can induce mutations in zebrafish. Using the synthetic modified crRNA:tracrRNA duplex instead of in vitro-transcribed single guide RNA, the low activity at some loci were dramatically improved in zebrafish. As far as we know, our work is the first report on the evaluation of ScCas9 in animals. Our work optimized ScCas9 as a new nuclease for targeting relaxed NNG PAMs for zebrafish genome editing, which will further improve genome editing in zebrafish.

## 1. Introduction

With the advantages of its high efficiency and simplicity of target design, the CRISPR/Cas9 system has been used to study various species, including zebrafish. The technique was simplified with the use of single guide RNA (sgRNA), containing a target site and chimeric CRISPR RNA (crRNA) and *trans*-activating crRNA (tracrRNA) sequences, that can direct Cas9 to the target site [1]. Target recognition only requires the presence of a protospacer adjacent motif (PAM) at the 3′ end of the target site.

The most widely used Cas9 protein is from *Streptococcus pyogenes* (SpCas9), which uses “NGG” as PAM sequence [2]. However, this specific PAM sequence may not be available near the target of interest, which limits the target sites that can be selected for gene editing applications commanding for high resolution target site positioning, such as efficient homology-directed repair mediated disease-associated mutations and base editing [3,4]. Additional Cas9 and Cpf1 variants with different PAM requirements have been discovered or engineered to diversify the range of targetable DNA sequences [5,6,7,8,9,10,11]. Although most of the variants exhibited robust editing use in human cells, plants, and many other model organisms [12,13], only limited Cas9 orthologs or variants were reported to have activity in zebrafish [14,15].

An orthologous Cas9 protein from *S. canis* (ScCas9) that shares 89.2% sequence similarity with SpCas9 has been identified and characterized, and is a promising candidate for genome editing in zebrafish [16]. ScCas9 recognizes minimal “NNG” PAM sequences and is capable of efficient genome editing in human cells and plants [16,17]. However, the editing ability of ScCas9 in zebrafish has not been tested.

Here, we evaluated the activity of the orthologous Cas9 protein ScCas9 in zebrafish. Through optimization, we showed that ScCas9 can edit the zebrafish genome with high targeting efficiency. This increases the frequency of available target sites, and expands the use of CRISPR/Cas9 in zebrafish by targeting previously inaccessible Cas9 sites in the genome.

## 2. Materials and Methods

### 2.1. Zebrafish Husbandry

Wild-type Tu line zebrafish were raised at 28.5 °C and embryos were staged according to description by Kimmel et al. [18].

### 2.2. Plasmid Construction

The full-length zebrafish codon-optimized bpNLS-ScCas9-bpNLS sequence was synthesized by GenScript, Nanjing, China and cloned into the pCS2+ vector. For protein expression, the ScCas9 coding sequence was subcloned into pET-28b vector. All cloning was done using ClonExpress II One Step Cloning Kit (Vazyme, Nanjing, China). The related sequence and primers can be found in Appendix A.

### 2.3. crRNA, tracrRNA, sgRNA and mRNA Synthesis

ScCas9 mRNA was in vitro transcribed from a *Not*I linearized ScCas9 vector using the SP6 mMESSAGE mMACHINE kit (Ambion, Carlsbad, CA, USA). SpCas9 mRNA was in vitro transcribed from an *XbaI* linearized zCas9 vector (Addgene, #46757) using the T3 mMESSAGE mMACHINE kit (Ambion, Carlsbad, CA, USA). All sgRNA templates in this study were synthesized using the cloning-independent sgRNA generation method [19], and sgRNAs were transcribed in vitro using the T7 MAXIscript kit (Ambion, Carlsbad, CA, USA). All in vitro-transcribed RNAs were purified using an RNeasy FFPE kit (QIAGEN, Dusseldorf, Germany) and quantified by NanoDrop 2000 (Thermo Fisher Scientific, Wilmington, NC, USA). All crRNAs and tracrRNAs were chemically synthesized by GenePharma, Shanghai, China or GenScript, Nanjing, China and dissolved in RNase-free water as a 25 µM stock solution in −80 °C. All primers and target sites are listed in Appendix A, respectively.

### 2.4. Sccas9 Expression and Purification

The ScCas9 protein was expressed in *Escherichia coli* strain BL21 Rosetta 2 (DE3). First, the transformed cultures were grown in 15 mL LB medium with 50 mg/L kanamycin at 180 rpm and 37 °C overnight. Starter cultures were then inoculated into 2 l LB medium that contained kanamycin and grown at 18 °C until A_600_ reached 0.6. The cultures were then induced with 0.5 mM IPTG and continued shaking at 200 rpm and 18 °C for 18 h. Cell pellets were harvested by multiple centrifugation rounds at 6850× *g* and 4 °C for 8 min, and resuspended in 30 mL lysis buffer that contained 50 mM NaH_2_PO_4_ with pH 8.0, 300 mM NaCl, 10 mM imidazole, 1 mM TCEP, 10% glycerol, lysozyme, 1 mM PMSF, and 0.1% Triton X-100. The cell suspension was lysed by repeated freezing and thawing, and then sonicated on ice for 1 h (40% peak intensity power, 6 s on, 10 s off). The cell lysate was centrifuged for 20 min at 13,500× *g* and 4 °C. The supernatant was added to 5 mL HisPur Ni-NTA Resin (QIAGEN, Dusseldorf, Germany) that was pre-equilibrated with 5 column volume (CV) lysis buffer. The protein-bound resin was washed with 5 CV wash buffer (20 mM HEPES, pH 7.4, 150 mM KCl, 1 mM TCEP, 10% glycerol, 0/20/40/60 mM imidazole). Protein was eluted with 2 mL elution buffer (20 mM HEPES pH 7.4, 150 mM KCl, 10% glycerol, 1 mM TCEP, 500 mM imidazole). All eluted fractions were visualized by SDS-PAGE with Instant-Bands (EZBiolab, Parsippany, NJ, USA), and dialyzed in SEC buffer (20 mM HEPES, pH 7.4, 150 mM KCl, 1 mM TCEP, 10% glycerol). After conducting dialysis treatment four times, the protein was concentrated with a 100 MWCO Amicon Ultra-15 mL Centrifugal Filter Unit (Millipore, Boston, MA, USA). Concentrated protein was confirmed by western blotting with His-tag antibody (GenScript, Nanjing, China) and stored at −80 °C.

### 2.5. Guide RNA: Cas9 Ribonucleoprotein (RNP) Complexes Preparation and Zebrafish Microinjection

In this study, all synthesized crRNA and tracrRNA stocks were mixed and annealed to form a stable chimeric duplex guide RNA (dgRNA) at a molar ratio of 1:1. To generate 5 µM in a 5 µL reaction system, 1 µL 25 µM sgRNA or dgRNA was incubated with 1 µL of 25 µM Cas9 stock in reaction buffer (100 mM NaCl, 50 mM Tris-HCl, 10 mM MgCl_2_, 1 mM DTT, pH 7.9) and 3 µL H_2_O at 37 °C for 15 min. One-cell stage zebrafish embryos were injected with 1 nL of a solution that contained an mRNA (300 ng/µL)/sgRNA (30 ng/µL) duplex or 5 µM RNP complex. After 2 days post-fertilization (dpf), injected embryos were collected for genotyping or imaging.

### 2.6. Mutations Detection

The mutagenesis efficiency of the target sites was assessed by TA cloning or T7 Endonuclease I (T7EI) assays. Briefly, genomic DNA was extracted from three pools of six arbitrarily collected embryos using the HotSHOT method [20]. Targeted genomic loci were amplified from genomic DNA and cloned into the pEASY-T1 vector (Transgene, Beijing, China) for Sanger sequencing using a minimum of 20 clones. In some cases, the amplified genomic DNA was assessed by T7 Endonuclease I Assay (NEB, Ipswich, UK) [21]. Briefly, the purified PCR products were annealed in NEB Buffer 2 with the following PCR program (95 °C, 5 min; 95–85 °C at −2 °C/s; 85–25 °C at −0.1 °C/s; hold at 4 °C) to form hybridized dsDNA. The hybridized dsDNA were then treated with 2 μL T7EI at 37 °C for 15 min in a reaction volume of 20 μL. Then, the digested samples were analyzed by electrophoresis through a 2% agarose gel. The band intensity was quantified using ImageJ 1.52a. Indel percentage was estimated by the formula: gene modification efficiency (test sample) = 1 − ((1-fraction cleaved) ^1/2^).

### 2.7. Imaging

Embryos were anesthetized with 0.03% tricaine (Sigma-Aldrich, Saint Louis, MO, USA) and mounted in 4% methylcellulose. All images were captured by a Zeiss Axio Imager Z1 microscope with the AxioCam MRc5 digital camera (Zeiss, Oberkochen, Germany) and processed by Adobe Photoshop CC 2018.

### 2.8. Founder and Stable Mutant Line Identification

Injected embryos were grown to adulthood and screened by pairwise outcrosses with wild-type fish. After 2 dpf, embryos from the progeny were collected as pools of 5 embryos/well and subjected to DNA extraction, PCR amplification, and Sanger sequencing. Sequence chromatograms were analyzed using Chromas. Germline transmission efficiency was then confirmed by re-breeding of several founders and sequencing of individual embryos (minimum of 24 embryos/founder).

### 2.9. Statistical Analysis

Statistical analysis was performed using GraphPad Prism 8. Significant differences (* *p* value < 0.05, ** *p* value < 0.01, *** *p* value < 0.001) from at least three independent experiments were determined by two-sided unpaired Student’s *t*-test.

## 3. Results

### 3.1. ScCas9 RNP Complexes Provide Robust Genome Editing in Zebrafish

To test the ScCas9 activity in zebrafish, we first synthesized a zebrafish codon that optimized ScCas9 that contained bpNLS sequences at both terminals; then, we cloned it into the pCS2+ vector (Figure 1A and Appendix A). 

To easily and efficiently test the ScCas9 activity, the *tyrosinase* (*tyr*) gene with a previously published sgRNA with a 5′-NGG-3′ PAM was chosen [22]. Tyrosinase encodes an enzyme that converts tyrosine into melanin, and a mutation in *tyr* results in loss of eye and body pigment. As a result, we can easily determine the *tyr* mutation efficiency according to the proportion of different pigment levels (Figure 1B).

Injection of the same amount of ScCas9 mRNA into the one-cell stage zebrafish embryos worked but greatly decreased the percentage of embryos displaying pigment loss compared with the SpCas9 mRNA group (Figure 1C). A previous study demonstrated that the SpCas9 protein can increase the indel frequency compared with SpCas9 mRNA in zebrafish [23]. Therefore, we purified recombinant ScCas9 protein using *Escherichia coli* expression system (Figure 1A and Appendix A). Injection of the ScCas9:sgRNA RNP complex into embryos increased the percentage of embryos displaying pigment loss relative to ScCas9 mRNA. Moreover, the activity of ScCas9 protein was comparable to that of SpCas9 mRNA (Figure 1C). Therefore, we used ScCas9 protein instead of ScCas9 mRNA in the following experiments.

### 3.2. ScCas9 Can Recognize NNG PAM in Zebrafish Genome

A previous study showed that only closely related Cas9 proteins can exchange their cognate dual-RNAs and still exert cleavage activity when using the Cas9 specific PAM [24]. Because the SpCas9 sgRNA scaffold is more universal and ScCas9 is a close ortholog of SpCas9, we wanted to determine whether the sgRNA scaffold is exchangeable (Figure 2A). To test the influence of sgRNA scaffold substitute in ScCas9, we designed a sgRNA targeting *rpl17* NAG PAM. T7EI results showed that the sgRNA scaffold substitution did not affect ScCas9 RNP activity in zebrafish (Figure 2B). Therefore, we used the SpCas9 sgRNA scaffold to synthesize ScCas9 sgRNA in the following experiments.

The powerful advantage of ScCas9 is its relaxed PAM (5′-NNG-3′) requirement [16]. Therefore, 18 targets with NAG, NCG, or NTG PAMs from six endogenous genes (*tyr*, *rpl17*, *rpl9*, *rpl31*, *ddx21*, *rps16*) were designed to evaluate the gene editing ability of ScCas9 in zebrafish (Appendix A). To determine the indel frequency induced by ScCas9 in zebrafish, targeted genomic loci were amplified from genomic DNA and then cloned for Sanger sequencing. At the seven NAG PAM target sites, indels were detected in all injected embryos with a frequency of 4.2–95.8%. Of the five NCG PAM target sites, ScCas9 had genome-editing ability at three targets and the editing efficiency was as high as 76.2%. Of the six NTG sites, only one locus had an indel mutation efficiency of 50% (Figure 2C and Appendix A). These results indicate that ScCas9 recognizes NNG PAM sites and has a preference for NAG PAM in zebrafish.

### 3.3. Chemical Modified dgRNP Can Significantly Improve ScCas9 Activity in Zebrafish Genome

A previous study showed that crRNAs with 2′-O-methyl 3′ phosphorothioate (MS)-modified nucleotides at both termini were exonuclease resistant while retaining gene editing activity in cell lines [25,26]. In addition, in vitro-assembled RNPs composed of Cas9 and crRNA:tracrRNA duplex (dgRNP) either with or without Alt-R modifications were more effective than sgRNA complexed with Cas9 protein (sgRNP) in zebrafish [27]. Therefore, we designed dgRNPs of several target sites across NNG PAMs to optimize the ScCas9 nuclease system.

Chemical MS modification was incorporated at two terminal nucleotides in both termini of crRNAs and tracrRNAs to evaluate their effects on efficacy compared with chemical synthesized crRNA:tracrRNA duplex and in vitro-transcribed sgRNAs (Figure 3A). Our results showed that the ScCas9 activities on *tyr*-NAG, *ddx21*-NAG, and *rpl9*-NCG were 12.5%, 47.6%, 14.3%, respectively, and increased to 57.5%, 73.1%, 15.0% with dgRNP, respectively. Interestingly, with chemical modified dgRNP (mdgRNP), the ScCas9 editing efficiency were dramatically improved to 88.0%, 88.9%, and 33.3% on the three loci, respectively (Figure 3B). These data demonstrated that the mdgRNPs (with 1.5–7-fold change efficiency) were significantly more effective than the sgRNPs in zebrafish (Figure 3B and Appendix A). 

## 4. Discussion

The PAM restriction limited the application of the CRISPR/Cas9 system in gene editing to some extent. Although researchers have made significant discoveries and engineering efforts to expand this restriction, only limited SpCas9 variants or orthologs were reported in zebrafish. In this study, we extensively determined the nuclease activity of ScCas9 on various NNG PAM target sites in zebrafish. 

In bacteria, it was first found that ScCas9 required NNGTT PAM; however, ScCas9 maintained comparable activity to relaxed NNG PAM in human cells [16]. Although 11 out of 18 target sites had indel efficiencies from 4–96%, ScCas9 was less active across NTG PAM and preferred NAG PAM in zebrafish (Figure 2C and Appendix A), which is consistent with the phenomenon observed in plants [28]. Moreover, the downstream sequence of the PAM had substantial effects on the ScCas9 activity, which led to ScCas9 preferring the more extended NAGBK (B = A, C, G; K = G, T) PAM than the shorter NNG PAM [29]. Interestingly, we found that the choice of PAM was not fully consistent with this rule for ScCas9 in zebrafish (Figure 2C).

To optimize the performance of the locus-dependent ScCas9, a new Sc++ was engineered with a novel PAM-interacting domain from two related ScCas9 orthologs and was suggested to be simultaneously more broad, efficient, and accurate than the original ScCas9 in human cells [30]. We tried this variant with a mRNA/sgRNA duplex on two target sites but did not find obvious improved efficiency (data not shown). Because of our limited data, further experiments with more targets should be conducted to more accurately evaluate Sc++ activity.

Because the ScCas9 and SpCas9 sequences are homologous and a previous result showed that sgRNA scaffold substitution did not affect the efficiency (Figure 2), we presumed that identical modifications from SpCas9 can be used to optimize the ScCas9 system [23,25,26]. Both the ScCas9 protein and chemically modified crRNAs increased the indel frequency similar to SpCas9. After the injected *rpl9*-NAG F_0_ embryos reached adulthood, we identified six of the eight individuals with indels by outcrossing, and Sanger sequencing of F_1_ embryo-specific PCR performed on a positive F_0_ embryo revealed comparable germline transmission efficiency (Appendix A). Therefore, our optimized ScCas9 system was more practical for zebrafish gene editing.

Until now, there was lack of single-base PAM-required CRISPR endonucleases in zebrafish. However, ScCas9, which is different from engineered SpCas9 variants such as xCas9 and SpCas9-NG [31,32], was first reported to target single-base PAM in zebrafish. ScCas9 substantially increases the editing scope in zebrafish, which might potentially expand the available base-editing tools and facilitate more precise homology-directed repair platforms.

## 5. Conclusions

In this study, we developed a procedure that uses the ScCas9 protein to efficiently introduce double-stranded break into the zebrafish genome. This system was optimized to target double-stranded DNA with the minimal NNG PAM and could expand the gene targeting scope in zebrafish; at a given site on the zebrafish genome, we now have more choices to select an appropriate nuclease for editing. This system should significantly expand CRISPR/Cas9 technology use in zebrafish.

## Figures and Tables

**Figure 1 cells-10-02099-f001:**
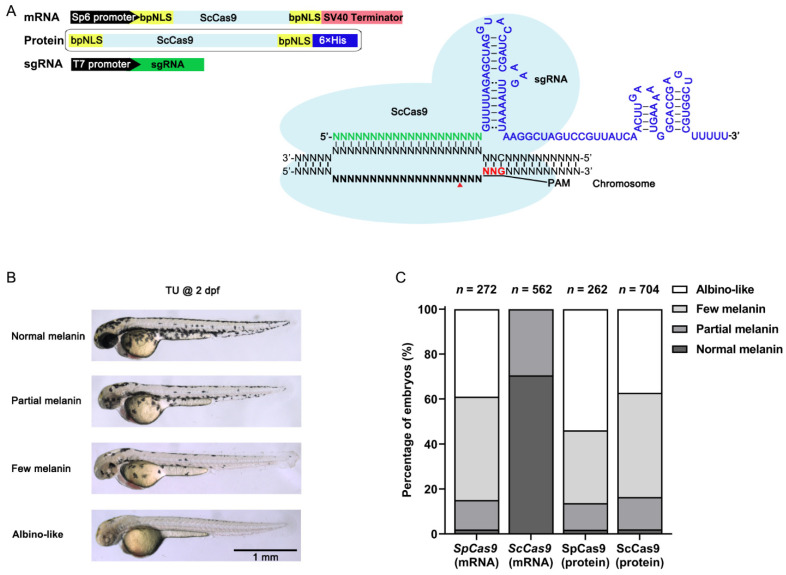
ScCas9 mRNA and protein induced *tyr* gene editing in zebrafish. (**A**), Schematic illustrating the ScCas9 system used in zebrafish. The ScCas9 system consists of two components, a dual bpNLS-tagged zebrafish codon-optimized ScCas9 protein and a sgRNA comprising a 20-nt seed sequence (green) infused with a sgRNA scaffold (blue). The ScCas9-bpNLS mRNA was transcribed in vitro from Sp6 promoter and sgRNA was transcribed from T7 promoter. ScCas9 protein tagged with bpNLS and His-tag was induced expression in vitro. Red triangle indicates the double stranded break sites induced by Cas9. bpNLS, bipartite nuclear localization signal; His, histidine tag. (**B**), Phenotypic evaluation of embryo pigment levels induced by targeting *tyr* using ScCas9 according to the amount of melanin. (**C**), Statistics of gene editing efficiency in the *tyr* target site induced by SpCas9 mRNA, ScCas9 mRNA, SpCas9 protein and ScCas9 protein.

**Figure 2 cells-10-02099-f002:**
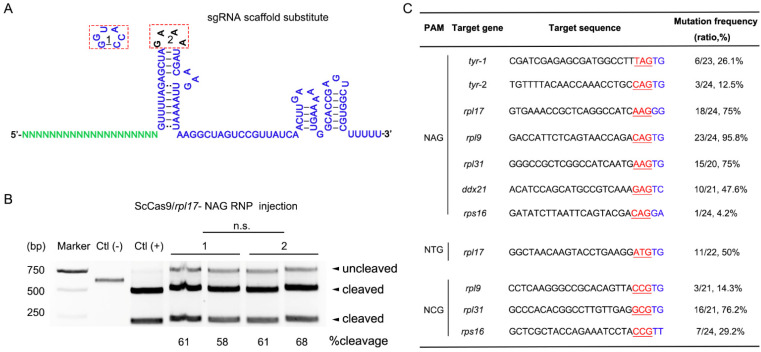
Genome editing in zebrafish using ScCas9. (**A**), Stem-loop differences between the ScCas9 original sgRNA tetraloop (1) and SpCas9 sgRNA tetraloop (2). (**B**), T7E1 assay of ScCas9/*rpl17*-NAG RNP-induced indel mutations. *BsrBI*-digested fragments of the purified PCR products were used as Ctl (+) and the undigested ones as Ctl (−). (**C**), Mutation frequency of ScCas9 at NNG PAM sites in zebrafish. The PAM is marked in red and underlined, and the downstream sequence of the PAM is marked in blue.

**Figure 3 cells-10-02099-f003:**
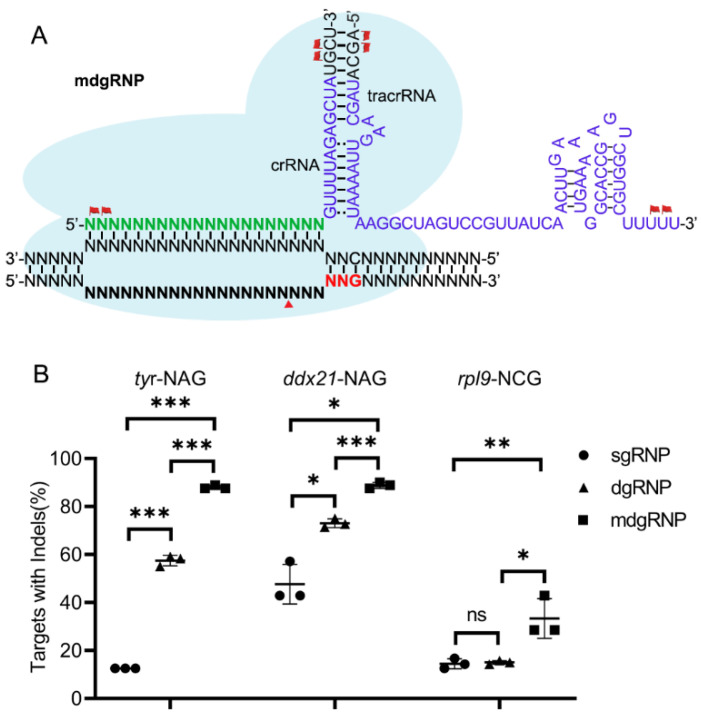
Chemically modified crRNA and tracrRNA duplex increased the ScCas9 efficiency. (**A**), Schematic illustration of the ScCas9 mdgRNP complex. The crRNA and tracrRNA scaffold with MS modified nucleotides (flag) at both termini coupled with the ScCas9 protein were loaded to the chromosome. (**B**), Statistics of gene editing efficiency induced by sgRNP, dgRNP and mdgRNP. ns, no significant difference; * *p* value < 0.05; ** *p* value < 0.01; *** *p* value < 0.001.

## Data Availability

The data that support the results of this study are available on reasonable request from the corresponding author.

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
