# Peer review of "Genome Editing in Zebrafish by ScCas9 Recognizing NNG PAM"

_cells, 2021, doi:10.3390/cells10082099_

Round 1
Reviewer 1 Report
General comments.
This study shows the application of CRISPR-ScCas9, which recognize the NNG PAM, to the zebrafish system by optimizing the previously published one.
High-efficiency genome editing was induced in a target-specific manner using chemically modified guide RNA.
It is a good study showing the application so that ScCas9 can be used as one of the genome editing tools in the future.
I'd like to suggest that this paper would be published in 'Cells' after minor correction.
minor comments
- In the Figure 1C, I wonder why the albino-like and Few melanin phenotype are not visible in the correction induced by ScCas9 mRNA.
It may confuse readers, so it would be good to show a graph organized by Cas9 expression method and phenotype, respectively.
- page 5 of 9, lane172-173: NAG PAM targeted editing still shows low activity(12.5%, 4.2%). authors didn't do sufficient experiment to
check the PAM specificity in zebrafish. So I'd like to suggest to change this sentence.
- page 7 of 9, lane234-235: authors didn't test the suitability of ScCas9 for HDR purpose. So I'd like to suggest to change this sentence.
Author Response
Please see the attachement.

Reviewer 2 Report
The most widely used SpCas9 requires NGG protospacer adjacent motif (PAM) sequence for binding to target DNA and producing double stranded DNA breaks. This sometimes limits target site selection in genome editing. In the present study, the authors tested ScCas9, which was shown to recognize NNG PAM and may be more convenient for target site selection, in inducing indels in the zebrafish genome. They showed that ScCas9 protein but not injected synthetic mRNA was as efficient as SpCas9 mRNA. Moreover, they examined the effects of chemical modifications of nucleotides on the editing efficiency of dgRNP in comparison with sgRNP. The observations made in this study may be helpful for genome editing in zebrafish. I have a few comments that need to be clarified.
1. By targeting the tyr locus, why injection of ScCas9 mRNA was not efficient in generating mutations, whereas injection of SpCas9 mRNA generated a high proportion of embryos without normal melanin? The authors should verify the translational efficiency of injected ScCas9 mRNA and provide an explanation. If injected ScCas9 mRNA lacks editing activity, this will limit its use.
2. The authors may also test the genome editing efficiency of ScCas9 mRNA on other target sites.
3. It is not clear whether the increase in genome editing efficiency by using dgRNP and chemical modifications was due to chemical modifications of nucleotides or simply due to the assembly of crRNA:tracrRNA duplex. The authors should also compare dgRNP without chemical modifications of nucleotides with sgRNP.
4. Line 85: what is the composition of the CV lysis buffer?
5. Line138: “greatly decreased the percentage of embryos…”, this sentence is not appropriate because injection of ScCas9 mRNA showed low editing efficiency but not decreased the editing efficiency of other Cas9.
6. Figure 1A, an RNA is transcribed from a promoter but should not contain the promoter sequence. The schematic that contains promoter sequences may be misleading.
7. Line 169: “Of the six NCG PAM target sites” and “Of the five NTG PAM target sites”. Table S3 shows five NCG target sites but six NTG target sites.
Author Response
Response to Reviewer 2 Comments
The most widely used SpCas9 requires NGG protospacer adjacent motif (PAM) sequence for binding to target DNA and producing double stranded DNA breaks. This sometimes limits target site selection in genome editing. In the present study, the authors tested ScCas9, which was shown to recognize NNG PAM and may be more convenient for target site selection, in inducing indels in the zebrafish genome. They showed that ScCas9 protein but not injected synthetic mRNA was as efficient as SpCas9 mRNA. Moreover, they examined the effects of chemical modifications of nucleotides on the editing efficiency of dgRNP in comparison with sgRNP. The observations made in this study may be helpful for genome editing in zebrafish. I have a few comments that need to be clarified.
Point 1: By targeting the tyr locus, why injection of ScCas9 mRNA was not efficient in generating mutations, whereas injection of SpCas9 mRNA generated a high proportion of embryos without normal melanin? The authors should verify the translational efficiency of injected ScCas9 mRNA and provide an explanation. If injected ScCas9 mRNA lacks editing activity, this will limit its use.

Response 1: Thanks for your questions. Previously study have shown that the chromatin-related factors might have a higher impact on eSpCas9 activity than SpCas9, which should be investigated thoroughly in future studies 1. As an orthologue of SpCas9, ScCas9 might be affected by its own ability to target heterogeneous chromatin too. Of Couse, that the protein translational speed of ScCas9 mRNA might be an important factor, which leads to the slow translated ScCas9 had weak ability to bind the complicated-state chromatin in late development period of zebrafish. In our study, we injected the same dose of mRNA (300 pg / embryo) since the equivalent molecular weight of ScCas9 and SpCas9. We found much more malformed embryos without improved editing efficiency when increasing the dose of mRNA injection, which limits the use of ScCas9 on mRNA level. Because early zebrafish fertilized eggs cleavage very fast (cells divide once every 15 min), and SpCas9 protein can increase the indel frequency compared with SpCas9 mRNA in zebrafish (line 142-144 in the revised manuscript), the form of ScCas9 protein directly supplied a fast mode of activity in vivo. Our study emphasized on the efficient gene editing method for ScCas9 in zebrafish but not for the mechanism. In fact, many other Cas9 orthologs and variants (xCas9, CjCas9) worked efficiently in cell line, but had no efficiency in zebrafish even though with RNP delivery (data not shown). Anyway, RNP delivery could also make the tool more widely use in zebrafish such as Cpf1 2-4.
- Jensen KT, Fløe L, Petersen TS, Huang J, Xu F, Bolund L, Luo Y, Lin L. Chromatin accessibility and guide sequence secondary structure affect CRISPR-Cas9 gene editing efficiency. FEBS Lett 2017, doi: 10.1002/1873-3468.12707.
- Moreno-Mateos MA, Fernandez JP, Rouet R, Vejnar CE, Lane MA, Mis E, Khokha MK, Doudna JA, Giraldez AJ. CRISPR-Cpf1 mediates efficient homology-directed repair and temperature-controlled genome editing. Nat Commun 2017, doi: 10.1038/s41467-017-01836-2.
- Fernandez JP, Vejnar CE, Giraldez AJ, Rouet R, Moreno-Mateos MA. Optimized CRISPR-Cpf1 system for genome editing in zebrafish. Methods 2018, doi: 10.1016/j.ymeth.2018.06.014.
- Meshalkina DA, Glushchenko AS, Kysil EV, Mizgirev IV, Frolov A. SpCas9- and LbCas12a-mediated DNA editing produce different gene knockout outcomes in zebrafish embryos. Genes (Basel) 2020, doi: 10.3390/genes11070740.
Point 2: The authors may also test the genome editing efficiency of ScCas9 mRNA on other target sites.
Response 2: We appreciate the suggestion. In fact, we had tested the genome editing efficiency of ScCas9 mRNA on other target sites in our study, such as rpl9-NAG, rpl9-NTG, rpl9-NCG, but the efficiency of ScCas9 mRNA was too low to detect. So we optimized the ScCas9 system for zebrafish gene editing.
Point 3: It is not clear whether the increase in genome editing efficiency by using dgRNP and chemical modifications was due to chemical modifications of nucleotides or simply due to the assembly of crRNA:tracrRNA duplex. The authors should also compare dgRNP without chemical modifications of nucleotides with sgRNP.
Response 3: Thanks for your comments. We have explained this chemical modified crRNA:tracrRNA duplex in line185-189 in the revised manuscript. That dgRNPs can be more effective than sgRNPs in zebrafish has been confirmed by previous study 5. So we didn’t compare dgRNPs with sgRNPs in our study anymore. Chemically modified crRNA:tracrRNA duplex was shown to functionally replace the natural guide RNA for efficient genome editing in human cells but not in zebrafish 6, 7. Therefore, we followed these two methods to optimize ScCas9 system.
- Hoshijima K, Jurynec MJ, Klatt Shaw D, Jacobi AM, Behlke MA, Grunwald DJ. Highly Efficient CRISPR-Cas9-based methods for generating deletion mutations and F0 embryos that lack gene function in zebrafish. Dev Cell 2019, doi: 10.1016/j.devcel.2019.10.004.
- Rahdar, M.; McMahon, M.A.; Prakash, T.P.; Swayze, E.E.; Bennett, C.F.; Cleveland, D.W. Synthetic CRISPR RNA-Cas9-guided genome editing in human cells. Proc Natl Acad Sci U S A 2015, doi:10.1073/pnas.1520883112.
- Hendel, A.; Bak, R.O.; Clark, J.T.; Kennedy, A.B.; Ryan, D.E.; Roy, S.; Steinfeld, I.; Lunstad, B.D.; Kaiser, R.J.; Wilkens, A.B.; et al. Chemically modified guide RNAs enhance CRISPR-Cas genome editing in human primary cells. Nat Biotechnol 2015, doi:10.1038/nbt.3290.
Point 4: Line 85: what is the composition of the CV lysis buffer?
Response 4: Thanks for reminding. There may be some ambiguities for this sentence. CV means column volume, and we added it in line 86-88 in the revised manuscript. We have given the composition of lysis buffer in line 80-84 in the revised manuscript.
Line 86-88 “The supernatant was added to 5 ml HisPur Ni-NTA Resin (QIAGEN) that was pre-equilibrated with 5 column volume (CV) lysis buffer.”
Point 5: Line138: “greatly decreased the percentage of embryos…”, this sentence is not appropriate because injection of ScCas9 mRNA showed low editing efficiency but not decreased the editing efficiency of other Cas9.
Response 5: Thanks for your comments. In figure 1C, we gave statistics of gene editing efficiency in the tyr target site induced by SpCas9 mRNA, ScCas9 mRNA, SpCas9 protein and ScCas9 protein, respectively. By comparing ScCas9 mRNA group and SpCas9 mRNA group, we did find injection of ScCas9 mRNA showed lower editing efficiency than SpCas9 mRNA.
Point 6: Figure 1A, an RNA is transcribed from a promoter but should not contain the promoter sequence. The schematic that contains promoter sequences may be misleading.
Response 6: Thanks for your comments. The schematic of figure 1A just illustrates the key elements for in vitro RNA transcription but not the final form of RNA we injected into embryos. Similar schematic was also appeared in other published literature 8-10. To make easily to understand the schematic of figure 1A, we have added some annotations in the figure legend in the revised manuscript (line 151-156 in the revised manuscript).
Line 151-156 “A, Schematic illustrating the ScCas9 system used in zebrafish. The ScCas9 system consists of two components, a dual bpNLS-tagged zebrafish codon-optimized ScCas9 protein and a sgRNA comprising a 20-nt seed sequence (green) infused with a sgRNA scaffold (blue). The ScCas9-bpNLS mRNA was transcribed in vitro from Sp6 promoter and sgRNA was transcribed from T7 promoter. ScCas9 protein tagged with bpNLS and His-tag was induced expression in vitro. Red triangle indicates the double stranded break sites induced by Cas9. bpNLS, bipartite nuclear localization signal; His, histidine tag.”
- Jao LE, Wente SR, Chen W. Efficient multiplex biallelic zebrafish genome editing using a CRISPR nuclease system. Proc Natl Acad Sci U S A 2013, doi: 10.1073/pnas.1308335110.
- Mali P, Yang L, Esvelt KM, Aach J, Guell M, DiCarlo JE, Norville JE, Church GM. RNA-guided human genome engineering via Cas9. Science 2013, doi: 10.1126/science.1232033.
- Zetsche B, Gootenberg JS, Abudayyeh OO, Slaymaker IM, Makarova KS, Essletzbichler P, Volz SE, Joung J, van der Oost J, Regev A, Koonin EV, Zhang F. Cpf1 is a single RNA-guided endonuclease of a class 2 CRISPR-Cas system. Cell 2015, doi: 10.1016/j.cell.2015.09.038.
Point 7: Line 169: “Of the six NCG PAM target sites” and “Of the five NTG PAM target sites”. Table S3 shows five NCG target sites but six NTG target sites.
Response 7: Done. We are sorry for making such mistakes. We have modified the description in line 174-177 in the revised manuscript.
Line 174-177 “Of the five NCG PAM target sites, ScCas9 had genome-editing ability at three targets and the editing efficiency was as high as 76.2%. Of the six NTG sites, only one locus had an indel mutation efficiency of 50% (Figure 2C and Figure S2).”
Reviewer 3 Report
Authors describe a study on a close ortholog of the widely adapted gene editing tool -RNA guided nuclease Streptococcus pyogenes Cas9 (SpyCas9). The Streptococcus canis (ScCas9) protein has been previously shown to recognize a less stringent NNG PAM in human and plant cells and authors use it to introduce indels and knock out genes in zebrafish. Two strategies of ScCas9 delivery into the genomes of zebrafish - embryo injection of ScCas9 and guide mRNA, and injection of assembled ScCas9:guide RNA RNP complexes - are compared and evaluated by phenotypic analysis. Authors then test the ScCas9 generated mutation frequency on various genomic targets with different PAM sequences to confirm the single nucleotide PAM stringency of ScCas9. The gene editing efficiency of the system is further optimized by using a duplex guide RNA molecule with chemically modified nucleotides.
In summary, authors claim that ScCas9 can be used to introduce double-stranded DNA breaks in zebrafish by targeting sequences adjacent to a NNG PAM.
However, some of the conclusions are unjustified, in addition authors uses baseless statements in the abstract and introduction. To improve the manuscript, I suggest following modifications:
Major points
- In the Abstracts and introduction authors states that, Spy Cas9 NGG PAM severely restricts the editable range of the zebrafish genome. However, ScCas9 PAM density is only 2× higher compared to SpyCas9. Is it so dramatically improving the situation? For example, for knock out mutations, which were tested in paper, precise position is not so important. Could the authors please be more specific and provide examples when it is not enough to target sequence with NGG PAM.
- It is erroneous to compare Spy mRNA with Sc protein delivery, and then draw conclusion that Sc is equally good as Spy. Please compare data from identical experiments. Is Sc is performing similarly as Spy, when RNP microinjection delivery is used?
- Line 173. In my opinion, conclusion "These results indicate that ScCas9 recognizes NNG PAM sites in zebrafish and is more suitable for editing NAG target sites in zebrafish" do not corresponds to results in figure 2C. Although some AG PAM targets worked very well, others like rps16 performed very poorly. I suggest concluding that ScCas9 has preference for NNG PAM, however different editing ratios shows that other PAM positions or other factors also might be important.
- Please note that in the Chatterjee et al used Spy Cas9 sgRNA for characterization of ScCas9, therefore 1 and 2 guide RNA designs showed in Figure 2A are from Spy Cas9. To be correct authors should compare repeat sequences and tracrRNA of Sc and Spy, and then design sgRNA of ScCas9.
- Dual guide RNA has not only chemical modifications, but also longer repeat-antirepeat duplex compared to sgRNA (12 vs 18 bp)? For direct comparison it would be appropriate to design sgRNA with longer repeat-antirepeat loop, since early experiments with SpyCas9 showed that length of sgRNA elements could be important for activity (Jinek, et al. 2014, Elife).
- It is not described, how activity of NHEJ was calculated from T7endo cleavage data. Please note, that the NHEJ efficiency do not correspond directly to T7endo cleavage efficiency and must be calculated using formula described by manufacturer of the reagent.
Minor points
1. According to known structures (Anders er al, 2014, Nature) PAM is recognized in dsDNA, but not in unwound DNA as represented in Figure 1A by authors. Please modify figure 1A accordingly?
2. Figures 2A and 3A, please use the same style for representation of the sane double stranded structure elements in sgRNA and tracrRNA.
Author Response
Please see the attachement.

Round 2
Reviewer 2 Report
A major issue raise in my previous review was not addressed in the revised manuscript.
"Point 3: It is not clear whether the increase in genome editing efficiency by using dgRNP and chemical modifications was due to chemical modifications of nucleotides or simply due to the assembly of crRNA:tracrRNA duplex".
This needs to be done by comparing the genome editing efficiency of dgRNP without and with chemical modifications.
Reviewer 3 Report
Authors took in to account most of reviewers comments and significantly improved the manuscript. I have only few minor comments:
- PAM is still recognised in single stranded DNA in fig. 1. Pleas correct it as you did with other figures.
- Authors should expand the T7 assay section in the methods to explain how intensity of NHEJ was calculated. Although authors explain it in the response, it should be coreccted in the methods section.
